# FORMALLY SPECIFYING THE HIGH-LEVEL BEHAVIOR OF LLM-BASED AGENTS

## ABSTRACT

LLM-based agents have recently emerged as promising tools for solving challenging problems without the need for task-specific finetuned models that can be expensive to procure. Currently, the design and implementation of such agents is ad hoc, as the wide variety of tasks that LLM-based agents may be applied to naturally means there can be no one-size-fits-all approach to agent design. In this work we aim to alleviate the difficulty of designing and implementing new agents by proposing a minimalistic, high-level generation framework that simplifies the process of building agents. The framework we introduce allows the user to specify desired agent behaviors in Linear Temporal Logic (LTL). The declarative LTL specification is then used to construct a constrained decoder that guarantees the LLM will produce an output exhibiting the desired behavior. By designing our framework in this way, we obtain several benefits, including the ability to enforce complex agent behavior, the ability to formally validate prompt examples, and the ability to seamlessly incorporate content-focused logical constraints into generation. In particular, our declarative approach, in which the desired behavior is simply described without concern for how it should be implemented or enforced, enables rapid design, implementation and experimentation with different LLM-based agents. We demonstrate how the proposed framework can be used to implement recent LLM-based agents, and show how the guardrails our approach provides can lead to improvements in agent performance. In addition, we release our code for general use.

## 1 INTRODUCTION

Many recent works (e.g., Brohan et al. (2023); Shen et al. (2023); Yao et al. (2022); Shinn et al. (2023)) have explored the use of large language models (LLMs) to drive the decision-making of intelligent, autonomous agents. Given a problem to solve, these LLM-based agents break the problem down into a sequence of steps, where each step involves either generating text or executing a tool (e.g., an API call Schick et al. (2023a)) which can supply new context to the agent. Importantly, while the order of steps to take is dictated by a high-level, prespecified behavior implemented by the user, the underlying LLM is still allowed a significant amount of flexibility in what it may produce. At each individual step, the outputs (i.e., the specific text or tool) are entirely determined by the LLM, thus allowing the agent to leverage the strong generative capabilities of LLMs while ensuring there are some guardrails to prevent aberrant behavior. Figure 1 provides an example of the popular ReACT Yao et al. (2022) agent framework, where an agent executes a loop that goes between generative steps (i.e., `Thought`, `Action`, `Action Input`) and tool execution steps (i.e., `Observation`).

Agent-based frameworks confer a number of benefits. First, these agent-based systems are significantly more performant than simpler systems utilizing standard prompting-based approaches (e.g., chain-of-thought Wei et al. (2022)). Second, because they allow the LLM to seamlessly integrate with external tools and incorporate the tool outputs into its context, agent-based approaches can address a much broader range of datasets and tasks (e.g., those involving web search Yang et al. (2018)). Lastly, by allowing end users to define an agent's behavior, the users are given more flexibility in tailoring very large language models to their particular needs than they would have otherwise.

```
Claim: Beautiful reached number two on the Billboard Hot 100 in 2003.?

Thought: I need to search the song Beautiful and find if it reached
number two on the Billboard Hot 100 in 2003.
Action: Search
Action Input: Beautiful
Observation: \"Beautiful\" is a song recorded by American singer
Christina Aguilera for her fourth studio album, Stripped (2002).

Thought: It does not mention Billboard, so I need to look up
\"Billboard Hot 100\" to find if it reached number two on it in 2003.
Action: Lookup
Action Input: Billboard Hot 100
Observation: (Result 1 / 3) The song peaked at number two on the
Billboard Hot 100 in the United States...

Final Thought: It only says the song peaked at number two on the
Billboard Hot 100, but not if it was in 2003. I am not sure if this
claim is true or not.
Answer: NOT ENOUGH INFORMATION
```

(a) Text output by ReACT agent

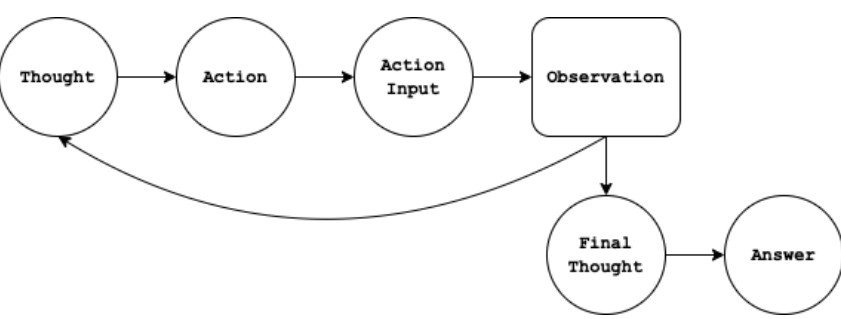

(b) Diagram of ReACT agent architecture

Figure 1: ReACT agent Yao et al. (2022) that alternates between generative steps (e.g., `Thought`, `Action`, `Action Input`) and tool execution steps (i.e., `Observation`)

Though LLM-based agents show much promise, there still remain challenges involved with their practical application. For instance, as each agent has its own strengths and weaknesses, it can be necessary to try a variety of different agents when approaching a problem. This can be a steep barrier to entry, as the lack of a standard framework for defining agents means that the end user must reimplement in code the exact behavior they wish for an agent to exhibit. In addition, even if a user finds that a particular agent type is well suited to their target task, the surface-level integration of current agent implementations with LLMs can mean running that agent will require several rounds of costly reprompting Xu et al. (2023). To address the aforementioned challenges, inspired by previous work on incorporating logic-based constraints into decoding Lu et al. (2021; 2022), we propose a declarative framework for formally specifying the high-level behavior of an LLM-based agent. To begin, our framework takes in an agent behavior specified in a simple fragment of linear temporal logic (LTL). The LTL specification is then used to define a constrained decoder that ensures the LLM-based agent executes steps in a way that conforms to the user's expectation. The specification thus serves as a type of contract for agent behavior, which provides straightforward opportunities for optimizing the generation process.

Our contributions in this work are as follows: (a) we introduce a declarative framework for defining LLM-based agents that is both lightweight and user-friendly, (b) we demonstrate the benefits of this framework, which includes the ability to enforce complex agent behavior, perform formal validation of prompt examples, and seamlessly incorporate content-focused logical constraints into generation, and (c) provide an analysis using three standard datasets (Hotpot QA Yang et al. (2018), Fever Thorne et al. (2018), and GSM8K Cobbe et al. (2021)) that demonstrates when the hard constraints imposed on LLM generation improve performance.

## 2 RELATED WORK

### 2.1 LLM-BASED AGENTS

The quest to develop agents that are capable of exhibiting intelligent behavior in real-world settings, without human intervention, has been a constant objective across the evaluation of AI Wooldridge & Jennings (1995); Weiss (1999). The recent advancement of LLMs unrolls new avenues of research to achieve this goal, where agents are backed with LLMs to perform complex tasks.

Various LLM-based agents targeting different tasks have been proposed, such as - WebAgent Gur et al. (2023) demonstrates the potential to create language-based agents capable of executing tasks on actual websites by adhering to natural language commands; Generative Agents Park et al. (2023) simulates believable human behavior; MetaGPT Hong et al. (2023) incorporates efficient human workflows as a meta-programming approach into LLM-based multi-agent collaboration; SayCan Ahn et al. (2022) illustrates the capability to use LLMs in a embodied agents. The success of these task-specific agents ignites the initiative to start open-source projects such as - AutoGPT [1], SuperAGI [2], and BabyAGI [3] that are focused on the objective of constructing self-sufficient agents capable of fulfilling users' requests.

### 2.2 TOOL AUGMENTED LLMS

In recent years, the development of large language models (LLMs) has made tremendous progress, and continues to drive research in prompt-based learning Wei et al. (2022); Khot et al. (2022) and instruction tuning Touvron et al. (2023); Gao et al. (2023b); Peng et al. (2023). While LLMs have demonstrated impressive performance, they are constrained by inherent limitations, with one of the primary drawbacks being their ability to utilize external tools. To address this limitation, there has been an increasing focus on the exploration of incorporating external tools into LLMs or creating tool-augmented LLMs.

Self-supervised or self-instructed learning is the leading methodology of augmenting tools into LLMs and we have witnessed multiple works in this direction such as ToolFormer Schick et al. (2023b); Gorilla Patil et al. (2023); TALM Parisi et al. (2022); etc. Through the integration of external tools, these augmented LLMs can achieve precise mathematical reasoning Cobbe et al. (2021); Thoppilan et al. (2022), access up-to-the-minute information with the assistance of web search engines Nakano et al. (2021), and harness domain-specific knowledge from external resources Yu et al. (2022). Additionally, some works use the Python interpreter to create complex programs, which helps them perform logical reasoning tasks better by accessing powerful computational resources Gao et al. (2023a); Chen et al. (2022).

### 2.3 CONSTRAINED DECODING

There has been a long line of constrained generation methods that modify the standard beam search decoding procedure at inference time, to incorporate constraints in the output. Lu et al. (2021; 2022); Hokamp & Liu (2017); Post & Vilar (2018). Anderson et al. (2017) proposes a constrained beam search algorithm that keeps track of constraints via a finite-state machine, and demonstrates its benefits on several image captioning tasks. Their method forces the inclusion of selected tag words in the output, and fixed, pre-trained word embeddings to facilitate vocabulary expansion to previously unseen tag words. Lu et al. (2021) proposes NEUROLOGIC DECODING, which enforces the satisfaction of lexical constraints (specified as any predicate logic formula having word inclusion and exclusion constraints) via adding a penalty term for constraint violation in the beam search decoding algorithm. Lu et al. (2022) improves on this by incorporating the $A^*$-search algorithm using lookahead heuristics at each decoding step. Bastan et al. (2023) builds on top of NEUROLOGIC DECODING by incorporating structural constraints that capture dependency parsing information. They demonstrate that the use of search algorithms, as well as structural constraints, improve performance over NEUROLOGIC on a variety of tasks such as lexical constrained generation, summarization, and machine translation.

---

[1] https://github.com/Significant-Gravitas/AutoGPT
[2] https://github.com/TransformerOptimus/SuperAGI
[3] https://github.com/yoheinakajima/babyagi

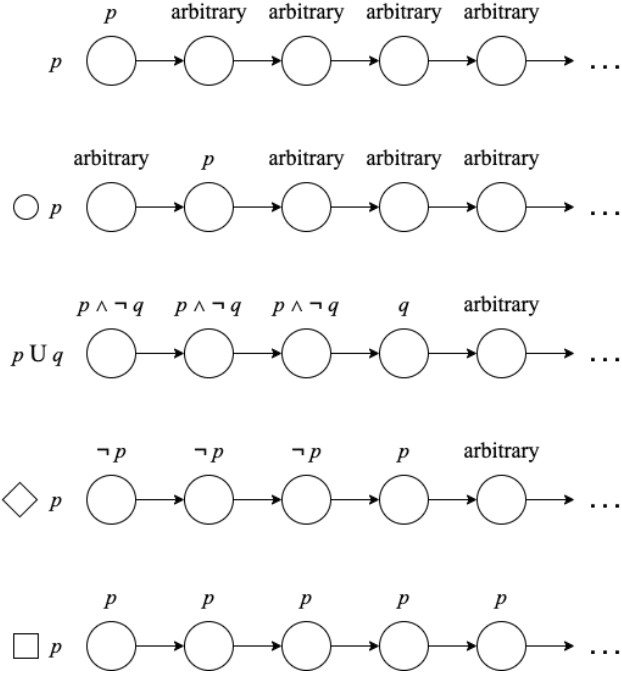

Figure 2: Examples of truth assignments over time with various LTL operators

Despite the significant progress in improving text generation using inference-type constraints, there remains a significant gap in the literature for improving generation from large-language models during inference, particularly for constraining the thought and action sequences as specified, for language models as agents. Thus, in this work we take the first step toward this by constraining agents to follow specifications for example, in ReACT Yao et al. (2022), Reflexion Shinn et al. (2023), chain-of-thought Wei et al. (2022) and chat-bot agents.

## 3 METHOD

In this section, we introduce our framework for designing and implementing autonomous agents that can interact with the environment to solve problems expressed in natural language. Our framework is intended to be lightweight (i.e., add as little additional overhead to LLM operation as is possible) and declarative (i.e., the user specifies the desired high-level behavior in terms of constraints without concern for how they should be implemented or enforced). To begin, we introduce linear temporal logic, which underpins our agent specification framework. Then, we provide a more formal definition of agents and how they are specified in our framework. Last, we describe how the specification framework is used to define a constrained decoder that controls what an agent can generate.

### 3.1 LINEAR TEMPORAL LOGIC

In this section, we provide a light overview of linear temporal logic (LTL), which is the key component to our agent specification framework. LTL is a modal temporal logic originally introduced for formal verification Pnueli (1977) that extends propositional logic with the temporal operators $\bigcirc$ (*next*) and $\mathcal{U}$ (*until*). The two operators have intuitive definitions, with $\bigcirc$ (*next*) being a unary operator that (informally) means a formula $\varphi$ must hold in the next time step, and $\mathcal{U}$ (*until*) being a binary operator that specifies a formula $\varphi_i$ must be true until $\varphi_j$ becomes true. LTL formulas are defined over a set of atomic propositions $\mathcal{P}$ with their syntax given by

$$\varphi ::= \text{true} \mid p \mid \neg\varphi \mid \varphi_1 \wedge \varphi_2 \mid \bigcirc \varphi \mid \varphi_1 \mathcal{U} \varphi_2 \qquad \text{where } p \in \mathcal{P}$$

An LTL formula is evaluated over an infinite sequence of observations, where each observation is a truth assignment over symbols in $\mathcal{P}$. Letting $\varphi$ be a LTL formula and $\sigma$ be the sequence of

observations $\sigma = \langle \sigma_1, \sigma_2, \ldots \rangle$, where each $\sigma_i$ can be considered the subset of $\mathcal{P}$ that is true at time $i$, then we write $\sigma \models \varphi$ (satisfies) when

$$
\begin{array}{llll}
\sigma & \models & \text{true} & \\
\sigma & \models & p & \text{iff} \quad p \in \sigma_0 \\
\sigma & \models & \neg \varphi & \text{iff} \quad \sigma \not\models \varphi \ (i.e., \sigma \models \varphi \ does \ not \ hold) \\
\sigma & \models & \varphi_1 \wedge \varphi_2 & \text{iff} \quad \sigma \models \varphi_1 \ \text{and} \ \sigma \models \varphi_2 \\
\sigma & \models & \bigcirc \varphi & \text{iff} \quad \sigma[1 \ldots] \models \varphi \\
\sigma & \models & \varphi_1 \, \mathcal{U} \, \varphi_2 & \text{iff} \quad \exists j \geq 0. \ \sigma[j \ldots] \models \varphi_2 \ \text{and} \ \sigma[i \ldots] \models \varphi_1, \quad \text{for all} \ 0 \leq i < j
\end{array}
$$

where $\sigma[i \ldots] = \langle \sigma_i, \ldots \rangle$ is the remaining sequence of observations following time step $i$. From the operators listed above, we can define additional propositional logic operators $\vee$ (*disjunction*), and $\rightarrow$ (*implication*) as well as temporal operators $\lozenge$ (*eventually*) and $\square$ (*always*)

$$
\begin{array}{lll}
\varphi_1 \vee \varphi_2 & := & \neg(\neg\varphi_1 \wedge \neg\varphi_2) \\
\varphi_1 \rightarrow \varphi_2 & := & \neg\varphi_1 \vee \varphi_2 \\
\lozenge \, \varphi & := & \text{true} \, \mathcal{U} \, \varphi \\
\square \, \varphi & := & \neg \, \lozenge \, \neg \varphi
\end{array}
$$

In addition, we also define a convenient shorthand notation for a chained sequence of *next* operators

$$
\bigcirc \langle \varphi_1, \varphi_2, \varphi_3, \ldots \rangle := \varphi_1 \wedge (\varphi_1 \rightarrow \bigcirc (\varphi_2 \wedge (\varphi_2 \rightarrow \bigcirc (\varphi_3 \wedge (\varphi_3 \rightarrow \bigcirc (\ldots))))))
$$

with this having the straightforward informal interpretation of "$\varphi_1$ then $\varphi_2$ then $\varphi_3$". In Figure 2, we provide a graphical depiction of the truth assignments over time for the above temporal operators.

In this work, we found that only allowing formulas to be those containing atomic propositions $p$, as well as operators $\rightarrow, \bigcirc, \square$, and $\mathcal{U}$ (i.e., we do *not* allow formulas to include $\neg, \wedge$, etc.) was sufficient to represent the range of existing agent architectures. We leave extending the set of operators (e.g., to include $\lozenge, \wedge$, etc.) to future work. For more details regarding LTL and its numerous applications, we direct the interested reader to Baier & Katoen (2008).

## 3.2 Specifying Agent Behavior

We model agents as generic transition systems, where a transition system is considered a tuple $\langle S, \delta, s_0, s_{end} \rangle$ consisting of a non-empty set of states $S$, a state transition function $\delta$, an initial state $s_0$, and a final state $s_{end}$. At each time step, the agent will receive a string from either an LLM or the environment [4], with the source of the string being determined by the particular state the agent is in. To define an agent and its underlying transition system, the user provides a specification consisting of 1) a list of states and their properties and 2) a desired behavior in the form of an LTL formula.

Figure 3 shows an example of a specification being provided in the format of a PDDL-style s-expression. In the specification, the `:states` list contains all possible states for the agent. Each state within the list must specify a prompt string (e.g., "Thought:" for the `Thought` state), which will serve as both an initial prompt for when the agent is in that state and as a signal to detect when a state transition occurs. By default it is assumed that the string received by an agent will be provided by the LLM. A user may override this by indicating the environment as the intended provider instead by using a special `:env-input` flag (e.g., shown in the `Observation` state).

The behavior of an agent is provided in the `:behavior` list. Referring back to the specification, we see an LTL formulation of the ReACT architecture (shown in Figure 1). In the formula, LTL operators (i.e., `next` and `until`) maintain the expected semantics (i.e., `next` interpreted as $\bigcirc \langle \ldots \rangle$ and `until` interpreted as $\mathcal{U}$). The LTL specification provides a constraint on the state transition function $\delta$, where only transitions that abide by the LTL formula are allowed (e.g., from the `Thought` state, the only possible transition is to the `Action` state). Lastly, unless otherwise specified, the initial and final states (i.e., $s_0$ and $s_{end}$, respectively) are always assumed to be the "first" and "last" states to appear in the LTL formula (in this case being `Thought` and `Answer`). Beyond ReACT, we also provide examples of how other prompting schemes are implemented in our framework, e.g., Shinn et al. (2023); Wei et al. (2022), in the Appendix.

---

[4] Here we refer to the "environment" as any provider of text that is *not* the LLM (e.g., external tools)

```
(define react-agent
  (:states
    (Thought (:text "Thought:"))
    (Action (:text "Action:"))
    (Action-Input (:text "Action Input:"))
    (Observation (:text "Observation:") (:flags :env-input))
    (Final-Thought (:text "Final Thought:"))
    (Answer (:text "Answer:"))
  )
  (:behavior
    (next
      (until
        (next Thought Action Action-Input Observation)
        Final-Thought)
      Answer)
  )
)
```

Figure 3: Specification for ReACT agent for the ReACT architecture shown in Figure 1(b)

## 3.3 CONSTRAINING AGENT BEHAVIOR

An LLM-based agent begins generation similarly to any other prompting-based approach. First, the input prompt, consisting of instructions and (optionally) a small number of examples, is concatenated with the text of the target problem and fed to the agent. Next, the agent is provided with the prompt text associated with the initial state $s_0$ (e.g., "Thought:" for the agent defined in Figure 3). At this point, the agent is considered to be in state $s_0$ and must now operate according to the behavior specified by its associated LTL formula. For the remainder of generation, the agent alternates between state transitions and text generation, with the agent terminating when it finishes generating text while in the final state, $s_{end}$.

State transitions are signaled when the underlying LLM produces the prompt text associated with any of its specification's states. Consider an agent in state $s_i$ where the next valid states are $\{s_j, \ldots\}$. We will refer to the text generated at this state as $t_i$, where $t_i$ may originate from either the LLM or the environment (e.g., the output of an API call). If $t_i$ ends with a prompt string associated with a valid state $s_j$, the agent has followed the specification successfully. Thus, the state transition is executed (i.e., the agent's state is set to be $s_j$) and the agent resumes generation from the next state. Alternatively, if $t_i$ does not end with a prompt string associated with a valid state, the agent enacts a state transition by picking a valid state $s_j$ arbitrarily and forcing its underlying LLM to decode to its associated prompt string (this is done with a decoder-side mask). This is now the same situation before, and so the state transition is executed and generation continues.

There are two nuances to consider for state transitions. First, when $t_i$ is produced by the environment, we do not expect $t_i$ to end with a particular state's prompt text, and thus the LLM is immediately called upon to enact a state transition when the agent receives $t_i$. Second, when $t_i$ is generated by the LLM, it may end by decoding an invalid state's prompt text (e.g., generating "Thought:" when it should generate "Action:"). For these cases, the incorrect state's prompt text is removed from $t_i$ and a state transition to a valid state is forcibly executed.

Lastly, we remark that one advantage of our system is the ease in which content constraints can be applied. In this work, we consider a content constraint to be a hard constraint on what the LLM is allowed to output in a particular state. For instance, a restriction on the set of allowed outputs in the tool-selection state to be only the names of valid tools (e.g., `Action` state in the ReACT framework). By allowing a user to specify what strings are acceptable within a particular state, the risk of certain types of hallucinations can be eliminated. Conveniently, the enforcement of a content constraint uses the same mechanism that is used to constrain state transitions.

```
Claim: Beautiful reached number two on the Billboard Hot 100 in 2003.?

Thought: I need to search the song Beautiful and find if it reached
number two on the Billboard Hot 100 in 2003.
Action: Search
Action Input: Beautiful
Observation: \"Beautiful\" is a song recorded by American singer
Christina Aguilera for her fourth studio album, Stripped (2002).

Thought: It does not mention Billboard, so I need to look up
\"Billboard Hot 100\" to find if it reached number two on it in 2003.
Action: Lookup
Action Input: Billboard Hot 100
Observation: (Result 1 / 3) The song peaked at number two on the
Billboard Hot 100 in the United States...

Final Thought: It only says the song peaked at number two on the
Billboard Hot 100, but not if it was in 2003. I am not sure if this
claim is true or not.
Answer: NOT ENOUGH INFORMATION
```

(a) Example of text output by ablated ReACT agent. Text that the agent is forced to decode is highlighted in green while text freely generated by the underlying LLM or the environment is highlighted in yellow

```
(define react-ablation-agent
  (:states
    (Thought (:text "Thought:"))
    (Observation (:text "Observation:") (:flags :env-input))
    (Final-Thought (:text "Final Thought:"))
  )
  (:behavior
    (until
      (next Thought Observation)
      Final-Thought)
  )
)
```

(b) Specification for ablated ReACT agent

Figure 4: ReACT agent with fewer constraints imposed on generation

## 4 EXPERIMENTS

We were interested to learn under what conditions the constraints imposed on generation were useful for improving agent performance. To explore this, we ran evaluations using the well-known ReACT Yao et al. (2022) architecture with a variety of different settings on three standard datasets: 1) GSM8K Cobbe et al. (2021), a mathematical reasoning dataset that tests the ability of a system to solve grade school math word problems, 2) HotpotQA Yang et al. (2018), a multi-hop question-answering dataset that requires reasoning over passages from Wikipedia, and 3) Fever Thorne et al. (2018), a claim verification dataset that requires assessing the validity of a claim against Wikipedia and predicting a label of "SUPPORTS", "REFUTES", or "NOT ENOUGH INFO".

Like Yao et al. (2022), we do not use the annotated Wikipedia passages and instead have the agent choose what terms to search in Wikipedia. Given the compute cost involved with running such a large number of experiments with large models and the number of questions involved in each dataset, for HotpotQA and Fever we selected a random subset of 500 questions from the development set for evaluation (however, for GSM8K we kept the full test set for evaluation). In addition, to control the generation size, we limited the number of ReACT loops to a maximum of 3 across all experiments.

### 4.1 SETUP

We hypothesized that the additional structure provided by a specification would be most useful for weaker LLMs. To determine if this was the case, we evaluated against an ablated version of the

| Model | Few-Shot $k$ | HotpotQA Acc. (%) | | Fever Acc. (%) | | GSM8K Acc. (%) | |
|---|---|---|---|---|---|---|---|
| | | ReACT Abl. | ReACT | ReACT Abl. | ReACT | ReACT Abl. | ReACT |
| MPT-7b | 0 | 0.2 | **1.6** | 10.2 | **29.8** | 2.6 | **3.9** |
| MPT-7b | 1 | 2.8 | **10.0** | 44.8 | **45.0** | 5.6 | **6.4** |
| MPT-7b | 2 | 5.8 | **8.8** | **53.4** | 53.0 | 5.1 | **6.1** |
| MPT-7b | 3 | 7.8 | **12.4** | **44.2** | 43.4 | 3.5 | **6.6** |
| MPT-30b | 0 | 3.8 | **7.2** | 34.0 | **36.4** | 12.1 | **14.4** |
| MPT-30b | 1 | 12.0 | **13.8** | 48.4 | **49.2** | 19.0 | **24.2** |
| MPT-30b | 2 | 14.4 | **15.2** | 45.0 | **45.2** | 23.4 | **28.1** |
| MPT-30b | 3 | 15.4 | **16.8** | 49.8 | **50.8** | 25.5 | **29.9** |
| SOTA ♣ | – | 67.5 | | 89.5 | | 97.0 | |
| ReACT ◇ | – | 27.4 | | 60.9 | | N/A | |

Table 1: HotpotQA Yang et al. (2018), Fever Thorne et al. (2018), and GSM8K Cobbe et al. (2021) results. "ReACT Abl." and "ReACT" refer to the partially and fully constrained agents. For HotpotQA and Fever, evaluation was performed on 500 question subsets of the development sets, while GSM8K evaluation was performed on the complete test set. For reference, we also provide the results for the supervised state-of-the-art ♣ (Zhu et al. (2021); Lewis et al. (2020); Zhou et al. (2023)) on all three datasets, and the results from the original ReACT ◇ which utilized PaLM-540B Chowdhery et al. (2022) as its underlying LLM

ReACT framework in experiments where we varied two hyperparameters of our system: 1) the number of examples $k$ provided in our few-shot prompts, which ranged from 0 to 3, and 2) the size of the underlying LLM in terms of number of parameters, where the model was an MPT instruction fine-tuned model Team (2023) with either 7 billion or 30 billion parameters.

Our implementation for both agents was built for the Huggingface Wolf et al. (2019) library, with state transitions being monitored and executed by a layer of code that lies on top of the built-in transformer models. Both systems had access to the same tools: 1) `Calculator` for GSM8K, which executes the formula provided as input and returns a number, 2) `Search` for HotpotQA and Fever, which returns the first 5 sentences from the Wikipedia page for the entity if it exists, or returns the 5 most similar entities with Wikipedia pages, 3) `Lookup` for HotpotQA and Fever, which returns the next sentence containing the input string in the last Wikipedia page searched for. To assess the utility of content constraints, we restricted the `Action` state to only output valid tools.

Our comparison system was a partially constrained agent (shown in Figure 4) without explicit state definitions for `Action`, `Action-Input`, and `Answer`. When the ablated specification was used, the agent was *not* forced to output the text associated with each of those states (i.e., in the `Thought` state, it was not forced to next output actions and their inputs). Instead, the only signal that those keywords should be part of the LLM output would come from the prompts given to the agent. For both the fully constrained and partially constrained agents, we took the final answer to be the text output after "Answer:", which should be found in the `Answer` and `Final-Thought` states for the standard and ablated agents, respectively. In the Appendix, we provide all prompts used to initialize both models (the prompts were the same for both agents).

## 5 RESULTS

In Table 1 we provide the results for all three datasets. The "ReACT Abl." and "ReACT" columns show results for the partially and fully constrained agents. In the tables, we see that the results largely follow our predictions, with the weakest models (i.e., the smaller models with fewer prompt examples) exhibiting the most significant differences between the normal and ablated agents. Notably, with only a few exceptions (i.e., $k \in \{1, 2\}$ for Fever), the ablated agent was always worse, which indicates the value of providing constraints during agent generation.

For both datasets, the most pronounced differences were observed in the zero-shot case (i.e., when the agent was given only the instructions for how to solve a problem with no examples). The 7b models had the widest gap on Fever, with the standard agent exhibiting a near 20% improvement

```
Question: Which movie is loosley based off the
Brother Grimm's \"Iron Henry\"?

Thought: I think Iron Man is the closest one.
Action: Lookup
Observation: Error in parsing action

Thought: I think Iron Man is the closest one.
Action: Search
Observation: Error in parsing action

Thought: I think Iron Man is the closest one.
Action: Lookup
Observation: Error in parsing action

Final Thought: Iron Man is the closest one
```

Figure 5: Example of an output where the ablated agent does not conform to the desired behavior

over the ablated agent. We attribute this to the smaller model's inability to follow instructions without an example, as the ablated agent made no prediction at all for 60% of all questions (i.e., did not predict one of the three possible labels for Fever). However, this appeared to only be an issue for the zero-shot case, as the ablated agent made predictions for all of Fever when a single prompt example was added.

The relative differences in performance between the two agents were more significant and more consistent for both HotpotQA and GSM8K than for Fever. We attribute this to Fever being a much simpler dataset, where the answer is only one of three possible options. Thus, whether the model got distracted or not, as long as it predicted one of the three answers it would have a good chance of answering correctly. In contrast, in both HotpotQA and GSM8K, the set of possible answers are almost entirely unconstrained. This helps to explain why the ablated agent marginally outperformed the unablated agent for $k \in \{1, 2\}$ for Fever.

Lastly, the number of parameters had a clear effect on the utility of constrained decoding. As evidenced by their performances in the zero-shot case, the 30b models were much more capable of following instructions. Still, that the 30b model in the ablated agent was outperformed uniformly by the main agent across all three datasets demonstrates that the use of constrained decoding may still be of some benefit as the datasets and instructions get more complex.

In Figure 5, we provide an example where the ablated agent does not follow the desired behavior (i.e., the behavior exemplified in the prompts given to the agent). As can be seen, the agent repeatedly skips the `Action-Input` step, which means the action (`Search` or `Lookup`) is executed with null inputs. In addition, the model terminates decoding after outputting for the `Final-Thought` state, which means there is no predicted answer.

## 6 CONCLUSION

In this work, we introduced a framework for defining agents using LTL in a task-agnostic, domain general fashion. The LTL formulation of an agent can be used to construct a constrained decoder, which controls the high-level behavior of an agent at runtime. We demonstrated the utility of our framework with experiments on three datasets, where we found that the hard constraints imposed on generation by our approach can lead to an increase in agent performance.

We discussed the several benefits and advantages of our approach. First and foremost, the introduction of a standard format for defining agents makes the process of implementing alternative agent types extremely straightforward. Second, because the LTL specification is provided prior to generation, it is straightforward to implement the monitor on top of the LLM generation code. Thus, many decoder-side optimizations are possible that can reduce the amount of redundant computation observed by prior works Xu et al. (2023); Wang et al. (2023). In addition, the logic-based approach makes it trivial to incorporate content constraints into decoding (as in Lu et al. (2021)).

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

# A APPENDIX

## A.1 HYPERPARAMETERS AND HARDWARE

There were not many hyperparameters to our approach beyond those listed in the experiments section, as our LLMs were only used for inference. The decoding strategy for our LLMs was set to greedy. In terms of hardware, our experimental setup utilized a HPC cluster with CPU and GPU machines running Red Hat Enterprise Linux release 8.7 (Ootpa). The GPU machines had 2 CPU cores and 100GB of RAM, and ran an NVIDIA A100 Tensor Core GPU with 80GB of GPU memory.

## A.2 AGENT PROMPTS

## A.3 AGENT DEFINITIONS

```
Determine if there is an observation that SUPPORTS or REFUTES
a Claim, or if there is NOT ENOUGH INFORMATION. You have
access to the following tools:
{{tool_descriptions}}
Use the following format:
Question:  the input question you must answer
Thought:  you should always think about what to do
Action:  the action to take, should be one of {{tool_labels}}
Action Input:  the input to the action
Observation:  the result of the action
...  (this Thought/Action/Action Input/Observation can repeat
N times)
Final Thought:  this is the last thought
Answer:  this should be one of [SUPPORTS, REFUTES, NOT ENOUGH
INFORMATION]

Here are some examples.

Claim:  Nikolaj Coster-Waldau worked with the Fox
Broadcasting Company.
Thought:  I need to look up Nikolaj Coster-Waldau and find if
he has worked with the Fox Broadcasting Company.
Action:  Lookup
Action Input:  Nikolaj Coster-Waldau
Observation:  Cannot use Lookup without using Search first!
Thought:  I need to search instead.
Action:  Search
Action Input:  Nikolaj Coster-Waldau
Observation:  Nikolaj William Coster-Waldau (born 27 July
1970) is a Danish actor and producer.  He graduated from
the Danish National School of Performing Arts in Copenhagen
in 1993,[1] and had his breakthrough role in Denmark with
the film Nightwatch (1994).  He played Jaime Lannister in
the HBO fantasy drama series Game of Thrones, for which he
received two Primetime Emmy Award nominations for Outstanding
Supporting Actor in a Drama Series..  Coster-Waldau has
appeared in numerous films in his native Denmark and
Scandinavia, including Headhunters (2011) and A Thousand
Times Good Night (2013).  In the U.S, his debut film role
was in the war film Black Hawk Down (2001), playing Medal of
Honor recipient Gary Gordon.[2] He then played a detective in
the short-lived Fox television series New Amsterdam (2008),
and appeared in the 2009 Fox television film Virtuality,
originally intended as a pilot.  Final Thought:  Because
he äppeared in the 2009 Fox television film Virtuality; he
should have worked with the Fox Broadcasting Company.
Answer:  SUPPORTS

Claim:
```

Figure 6: Fever Thorne et al. (2018) ReACT prompt with $k = 1$ few-shot examples

```
Answer the following questions as best you can.  You have
access to the following tools:
{{tool_descriptions}}
Use the following format:
Question:  the input question you must answer
Thought:  you should always think about what to do
Action:  the action to take, should be one of {{tool_labels}}
Action Input:  the input to the action
Observation:  the result of the action
...  (this Thought/Action/Action Input/Observation can repeat
N times)
Final Thought:  this is the last thought
Answer:  this should be one of [SUPPORTS, REFUTES, NOT ENOUGH
INFORMATION]

Here are some examples.

Question:  What is the elevation range for the area that the
eastern sector of the Colorado orogeny extends into?
Thought:  I need to lookup Colorado orogeny, find the area
that the eastern sector of the Colorado orogeny extends into,
then find the elevation range of the area.
Action:  Lookup
Action Input:  Colorado orogeny
Observation:  Cannot use Lookup without using Search first!
Thought:  I need to search instead.
Action:  Search
Action Input:  Colorado orogeny
Observation:  The Colorado orogeny was an episode of mountain
building (an orogeny) in Colorado and surrounding areas.
Thought:  It does not mention the eastern sector.  So I need
to look up eastern sector.
Action:  Lookup
Action Input:  eastern sector
Observation:  (Result 1 / 1) The eastern sector extends into
the High Plains and is called the Central Plains orogeny.
Thought:  The eastern sector of Colorado orogeny extends into
the High Plains.  So I need to search High Plains and find
its elevation range.
Action:  Search
Action Input:  High Plains
Observation:  High Plains refers to one of two distinct land
regions:
Thought:  I need to instead search High Plains (United
States).
Action:  Search
Action Input:  High Plains (United States)
Observation:  The High Plains are a subregion of the Great
Plains.  From east to west, the High Plains rise in elevation
from around 1,800 to 7,000 ft (550 to 2,130 m).[3]
Final Thought:  High Plains rise in elevation from around
1,800 to 7,000 ft, so the answer is 1,800 to 7,000 ft.
Answer:  1,800 to 7,000 ft

Question:
```

Figure 7: HotpotQA Yang et al. (2018) ReACT prompt with $k = 1$ few-shot examples

```
Answer the following questions as best you can.  You have
access to the following tools:
{{tool_descriptions}}
Use the following format:
Question:  the input question you must answer
Thought:  you should always think about what to do
Action:  the action to take, should be one of {{tool_labels}}
Action Input:  the input to the action
Observation:  the result of the action
...  (this Thought/Action/Action Input/Observation can repeat
N times)
Final Thought:  this is the last thought
Answer:  this should be one of [SUPPORTS, REFUTES, NOT ENOUGH
INFORMATION]

Here are some examples.

Question:  Natalia sold clips to 48 of her friends in April,
and then she sold half as many clips in May.  How many clips
did Natalia sell altogether in April and May?
Thought:  I need to use the number of clips in April to
determine the number of clips sold in May.  In May, Natalia
sold half of the amount of clips that she did in April.
Action:  Calculator
Action Input:  48 / 2
Observation:  24
Thought:  Now I need to add the number of clips I calculated
for May with the number of clips she sold in April
Action:  Calculator
Action Input:  24 + 48
Observation:  72
Final Thought:  So Natalia sold 72 clips altogether in April
and May.
Answer:  72

Question:
```

Figure 8: GSM8K Cobbe et al. (2021) ReACT prompt with $k = 1$ few-shot examples

```
(define react-agent
  (:states
    (Thought (:text "Thought:"))
    (Action (:text "Action:"))
    (Action-Input (:text "Action Input:"))
    (Observation (:text "Observation:") (:flags :env-input))
    (Final-Thought (:text "Final Thought:"))
    (Answer (:text "Answer:"))
  )
  (:behavior
    (next
      (until
        (next Thought Action Action-Input Observation)
        Final-Thought)
      Answer)
  )
)
```

Figure 9: Specification for ReACT agent

```
(define chain-of-thought-agent
  (:states
    (Thought (:text "Let's think step by step."))
    (Answer (:text "Answer:"))
  )
  (:behavior
    (next Thought Answer)
  )
)
```

Figure 10: Specification for Chain-of-Thought agent

```
(define reflexion-agent
  (:states
    (Thought (:text "Thought:"))
    (Action (:text "Action:"))
    (Action-Input (:text "Action Input:"))
    (Observation (:text "Observation:") (:flags :env-input))
    (Final-Thought (:text "Final Thought:"))
    (Answer (:text "Answer:"))
    (Evaluator (:text "Evaluation:") (:flags :env-input))
    (Reflection (:text "Reflection:"))
    (Finish (:text "Generation complete!"))
  )
  (:behavior
    (until
      (next
        (until
          (next Thought Action Action-Input Observation)
          Final-Thought)
        Answer
        Evaluator
        Reflection)
      Finish)
  )
)
```

Figure 11: Specification for Reflexion agent

```
(define chat-bot-agent
  (:states
    (Chat-Bot (:text "Chat Bot:"))
    (User (:text "User:") (:flags :env-input))
  )
  (:behavior
    (always
      (next Chat-Bot User)
    )
  )
)
```

Figure 12: Specification for Chat-bot agent

