# OpenReview forum: "Formally Specifying the High-Level Behavior of LLM-Based Agents"
_ICLR.cc/2024/Conference — ICLR 2024 Conference Withdrawn Submission_

### Official Review · Reviewer_nTyW · 2023-10-30

**Soundness:** 3 good
**Presentation:** 3 good
**Contribution:** 2 fair
**Rating:** 5
**Confidence:** 4

**Summary:**

For the purpose of designing and putting LLM-based agents into practice, the authors suggest a simple, high-level generation framework. In order to build a constrained decoder, the framework enables users to specify desired agent behaviors in Linear Temporal Logic (LTL). The method enables complex agent behavior, prompt example validation, and logical constraints. The experimental evaluation with various LLM-based agents are made possible by the declarative approach.

**Strengths:**

+ The paper address an important problem
+ It is easy to follow

**Weaknesses:**

- The novelty and the contributions of the paper are unclear
- The performance of the method seems limited

**Questions:**

This work proposes a lightweight framework for designing and implementing LLM-based agents. The framework allows users to specify desired agent behaviors in LTL. To me, the biggest problem of the paper is the novelty and the contributions. The introduction and related work part do not give an in-depth comparison with existing work like
- Silver, Tom, et al. "Generalized Planning in PDDL Domains with Pretrained Large Language Models." arXiv preprint arXiv:2305.11014 (2023).
- Sumers, Theodore, et al. "Cognitive architectures for language agents." arXiv preprint arXiv:2309.02427 (2023).

If the simple idea could be demonstrated more on the SOTA model, like PALM, I think it would make the result more persuasive.

Minor issues:

1. Section 3.2
> Figure 3 shows an example of a specification being provided in the format of a PDDL-style expression.

You should explain what is PDDL.

2. Section 3.3. I think the authors should explain the advantage of constraining agent behavior more. Also, tell the readers what else they can do with the proposed method.

3. Section 4: The comparison with ReACT Abl. is limited. The authors could compare the methods with different prompt engineering.

---

### Official Review · Reviewer_dXvQ · 2023-10-31

**Soundness:** 2 fair
**Presentation:** 2 fair
**Contribution:** 2 fair
**Rating:** 3
**Confidence:** 4

**Summary:**

The paper presents a framework for declarative design of LLM-based agents. In particular, the user/designer expresses the required behavior of the agent using a Linear Temporal Logic (LTL) specification. The agent is contrained to follow such specifications via constrained decoding of LLM outputs. To evaluate the effectiveness of the framework, ReACT-style agents are constructed using the declarative LTL approach and evaluated on three datasets. The constrained ReACT agents outperform the unconstrained ones. Moreover, the authors also provide example LTL specifications for other types of agents, namely, a Chain-of-Thought agent, a Reflexion agent, and a simple Chatbot agent.

**Strengths:**

1. I think it is an extremely interesting idea to use LTL specifications for constraining the behavior of not only LLM-based agents but also of LLMs in general. While there is a growing body of work on constrained decoding for LLMs, using LTL to express these constraints has not been explored and I believe this is a fruitful direction to explore.

2. The fact that the presented framework can be used to express a number of LLM-based agents (ReACT, CoT, Reflexion, Chatbots) suggests the generality of the approach.

**Weaknesses:**

The primary weakness of the paper is the sparsity and lack of precision regarding the technical details about the framework. I list my concerns below:

1. The transition system based formalization of the agent behavior (Section 3.2) is imprecise. What is the precise notion of a state? Does a state include a prompt string (such as "Thought:") along with the generated string? Does a state always need to start with a special string such as "Thought"? Also, to use LTL, each state needs to be associated with corresponding propositions. Such a mapping between states to propositions is never formally defined.

2. How is the set of next valid states from a current state calculated in general? The problem of determining the next set of valid states seems closely related to runtime monitoring of LTL specifications (see [1]). Building such monitors requires sophisticated techniques, so I find it concerning that there is no discussion about this in the paper.

3. Related to the previous question, is there ever a need to backtrack when enforcing an LTL specification? For instance, can it ever be the case that in a state $s_i$, it may seem there are multiple possible valid next states $s_{j1}, s_{j2}, ..., s_{jN}$ but later in the sequence one realizes that some of these states were not actually not valid and the agent needs to backtrack?

3. How is the LTL specification actually enforced? Consider the example of the ReACT agent. In a typical implementation, a *single* call to an LLM generates a response that includes "Thought", "Action", and "Action Input" states (for instance, see implementation of ReACT [here](https://github.com/ysymyth/ReAct)). But to enforce the LTL specification, one would need to constantly monitor the LLM output, i.e., as each token is produced by the LLM. How is this token-level monitoring implemented? If the framework does not use such token-level monitoring, then what is the precise mechanism used and is such mechanism generalizable to any LTL specification? Section 3.3 does not provide sufficient details.

4. It seems like the framework operates at a level of granularity that is higher than token-level granularity. What is precisely this level of granularity? Why can't existing constrained decoding approaches be used?

5. Although I believe that such techniques based on logically constraining the behavior of LLMs are very promising, the empirical results in the paper do not make a strong case for the same.

6. There is a lot emerging literature on constrained decoding with respect to logical constraints that is not cited. For instance, [2] and [3]

[1] Andreas Bauer, Martin Leucker, and Christian Schallhart. 2011. Runtime Verification for LTL and TLTL. ACM Trans. Softw. Eng. Methodol. 20, 4, Article 14 (September 2011), 64 pages. https://doi.org/10.1145/2000799.2000800

[2] Honghua Zhang, Meihua Dang, Nanyun Peng, and Guy Van Den Broeck. 2023. Tractable control for autoregressive language generation. In Proceedings of the 40th International Conference on Machine Learning (ICML'23), Vol. 202. JMLR.org, Article 1716, 40932–40945.

[3] Luca Beurer-Kellner, Marc Fischer, and Martin Vechev. 2023. Prompting Is Programming: A Query Language for Large Language Models. Proc. ACM Program. Lang. 7, PLDI, Article 186 (June 2023), 24 pages. https://doi.org/10.1145/3591300

**Questions:**

See the **Weaknesses** section

---

### Official Review · Reviewer_34dE · 2023-10-31

**Soundness:** 3 good
**Presentation:** 4 excellent
**Contribution:** 2 fair
**Rating:** 6
**Confidence:** 4

**Summary:**

- The paper introduces a linear temporal logic-based framework for specifying and implementing the architectures of large language model-based agents, which formalises and generalises existing LLM agent architectures.

- To specify an architecture, we write down the set of states the LLM agent can be in (e.g. thinking, acting) and use LTL to specify which sequences of states are valid. State transitions occur when the language model outputs a state signifier (e.g. THOUGHT:, ACTION:).  If the state signifier corresponds to a valid transition, the transition occurs. Otherwise a transition to a valid state is forcibly executed using a constrained decoder, which restricts the output of the model to the set of valid state signifiers.

- The key contribution of the paper is the pipeline which allows new agent architectures to be implemented easy and quickly.

- The authors carried out experiments to asses how useful it is to constrain the state transitions using LTL, compared to simply providing examples of acceptable state trajectories, or providing instructions with no examples.

- The constraints most strongly improved performance for small models given few examples, and the difference dropped off somewhat as the size of the model and number of examples increased.

**Strengths:**

- Originality: To my knowledge this is the first attempt at an overarching formal framework for specifying the architectures of LLM-based agents.

- Quality
        - Formal logic, and in particular LTL, are well-suited to specifying the architecture of LLM agents.
        - The pipeline is simple and intuitive, so it seems extremely easy to use and good for fast iteration of architectures.
        - The pipeline is expressive enough to capture the architecture of at least a few popular LLM agents, along with chain of thought.
        - The framework seems like a useful way to organise different architectures, and was somewhat conceptually clarifying for me as to the relationship between e.g. chain-of-thought and LLM agents.
        - The use of constrained decoding to enforce valid transitions seems appropriate

- Clarity
        - The paper is generally extremely clearly-written and well-organised. I felt I was able to immediately understand its core contributions.

- Significance
        - The paper opens the door to future work on specification pipelines for LLM agent architectures.

**Weaknesses:**

- I think the experimental results slightly undermine the significance of the paper's contribution. The performance improvement obtained using the pipeline made decreased with the size of the LLM. Since there are much larger LLMs than the ones used in these experiments, and we can expect to generally be working with larger models as time goes on, it seems constraining models using LTL may not be especially useful for improving performance - LLMs understand what trajectories of states are valid after seeing a few examples.

 - (Then again, there are surely situations where for the sake of safety rather than average performance we would like to fully constrain LLM agents - perhaps this is an alternative angle on the contribution.)

- The PDDL-style s-expression syntax used to actually specify the architectures is not defined. It can be deduced from knowledge of LTL and PDDL, but why leave such an important part of the paper to be deduced by the reader?

 - Only three examples of LLM agent architectures expressed in the framework are given, one of which (chain of thought is not a central example of an agent architecture. I am therefore unable to evaluate whether the framework is expressive enough to encompass most or all popular LLM agents, or just a few. This should be fairly easy to fix by adding more examples to the appendix, and commenting on how general the framework is.

- This is only a weak suggestion, but I felt that constrained decoding could be a bit more explicitly explained in the context of the paper, perhaps by walking through an example of its use in the pipeline.

**Questions:**

- Does the framework cover popular LLM agents such as AutoGPT, SuperAGI, and BabyAGI?

- Suggestions: Explicitly explain the syntax used to specify agents.

---

### Official Review · Reviewer_a6kU · 2023-11-01

**Soundness:** 1 poor
**Presentation:** 2 fair
**Contribution:** 1 poor
**Rating:** 3
**Confidence:** 4

**Summary:**

The paper discuss a reformulation of ReACT in the Linear Temporal Logic (LTL) framework for a more formal description. The authors first introduce the background, traditional definitions, and symbols for LTL. Then, they discuss on the connection between LTL and agent operation mechanism in the context of LLM, and present a technical suggestion to do fully constrained decoding at the beginning of certain operation (e.g., "Thought", "Action", "Final Thought", and related action space etc.). In experiments, they compared ReACT with and without such constraints on MPT-7B and 30B and three datasets, and show some improvements of the constrained setting compared to the unconstrained one.

**Strengths:**

1. The authors endeavor to present a more rigorous formulation of agent prompting in the context of LLM, which I appreciate as practices in this domain are still quite experimental and require standardization.
2. The paper has fair writing, which introduces the concept of LTL clearly.

**Weaknesses:**

1. Lack of Solid Contribution: while the concept of LTL is interesting, it is not an original idea of this paper. Further, the application of LTL in this paper seems unnecessary. The main technical proposal in this work--constrained decoding--has been a well-established practice in improving LM's structured generation. It has very weak connection with the LTL's formulation and so many logical symbols introduced, which occupies more than half of the method content.
2. Lack of Strong Experiment Results: all experiments are done with MPT-7B and 30B in in-context learning setting, which are insufficient in either model sizes or types. Some APIs that support prefix assigning should be considered, such as text-davinci-003. And more baselines and ablations (instead of only one) should be compared and presented.

**Questions:**

See above weakness.

---

### Official Review · Reviewer_M2DU · 2023-11-02

**Soundness:** 3 good
**Presentation:** 3 good
**Contribution:** 2 fair
**Rating:** 3
**Confidence:** 4

**Summary:**

This paper presents a high-level generation framework aimed at simplifying the design and implementation of new LLM-based agents. The framework allows users to specify desired agent behaviors using Linear Temporal Logic (LTL). It leverages a declarative approach to enable rapid design, implementation, and experimentation with different LLM-based agents based on LTL. The framework ensures the production of agent behaviors aligned with specified requirements and incorporates logical constraints into response generation. The experiment showcases the application of this framework in implementing LLM-based agents based on LTL and highlights its efficacy in enhancing agent performance.

**Strengths:**

* The declarative approach simplifies behavior specification, and the constrained decoder ensures accurate alignment of generated output with desired behaviors.

* The paper sets up a new way based on formal logic to control the high-level behavior of an LLM-based agent at runtime.

* The experiment results demonstrate the utility of the framework on three datasets and show that hard constraints imposed on generation can lead to an increase in agent performance.

**Weaknesses:**

* The paper asserts the capability to formally validate prompt examples by the proposed approach. However, I was unable to locate evidence supporting this claim.

* The experiment conducted falls short of demonstrating the value of integrating Linear Temporal Logic (LTL) specifications with Language Learning Models (LLMs). The evaluation exclusively focuses on the ReACT agent framework, where the agent operates within a loop involving generative and tool execution steps, resulting in behavior represented as a sequence of Thought, Action, Action Input, and Observation. A finite state transition system or straightforward rule-based systems could serve as monitors on the LLM generation code to impose the hard constraints. The necessity of LTL specifications remains unclear. Strengthening the paper would involve a case study featuring an agent with more complex temporal behaviors interacting with its environment to demonstrate the significance of LTL integration.

While I believe in the importance of enforcing hard constraints to monitor the Language Learning Models (LLM)-based agents' output, further experiments focusing on the efficacy and necessity of integrating Linear Temporal Logic (LTL) specifications is imperative to justify publication.

**Questions:**

Can you provide more details about your tool's capacity for formal validation of prompt examples? Has there been any evaluation conducted to assess its effectiveness in this regard?